# A Multivariant Surrogate Neutralization Assay Identifies Variant-Specific Neutralizing Antibody Profiles in Primary SARS-CoV-2 Omicron Infection

**DOI:** 10.3390/diagnostics13132278

**Published:** 2023-07-05

**Authors:** David Niklas Springer, Marianna Traugott, Elisabeth Reuberger, Klaus Benjamin Kothbauer, Christian Borsodi, Michelle Nägeli, Theresa Oelschlägel, Hasan Kelani, Oliver Lammel, Josef Deutsch, Elisabeth Puchhammer-Stöckl, Eva Höltl, Judith Helene Aberle, Karin Stiasny, Lukas Weseslindtner

**Affiliations:** 1Center for Virology, Medical University of Vienna, 1090 Vienna, Austria; david.springer@meduniwien.ac.at (D.N.S.); elisabeth.reuberger@meduniwien.ac.at (E.R.);; 24th Medical Department, Clinic Favoriten, Kaiser-Franz-Josef Hospital, 1100 Vienna, Austria; marianna.traugott@gesundheitsverbund.at (M.T.);; 3Independent Researcher, 8972 Ramsau am Dachstein, Austria; 4Independent Researcher, 9100 Völkermarkt, Austria; 5Center for Public Health, Medical University of Vienna, 1090 Vienna, Austria

**Keywords:** SARS-CoV-2, Omicron, antibodies, neutralization, surrogate assay, immunoassay

## Abstract

Primary infection with the Omicron variant of Severe Acute Respiratory Syndrome Corona Virus 2 (SARS-CoV-2) can be serologically identified with distinct profiles of neutralizing antibodies (nAbs), as indicated by high titers against the Omicron variant and low titers against the ancestral wild-type (WT). Here, we evaluated whether a novel surrogate virus neutralization assay (sVNT) that simultaneously quantifies the binding inhibition of angiotensin-converting enzyme 2 (ACE2) to the proteins of the WT- and Omicron-specific receptor-binding domains (RBDs) can identify nAb profiles after primary Omicron infection with accuracy similar to that of variant-specific live-virus neutralization tests (NTs). Therefore, we comparatively tested 205 samples from individuals after primary infection with the Omicron variant and the WT, and vaccinated subjects with or without Omicron breakthrough infections. Indeed, variant-specific RBD-ACE2 binding inhibition levels significantly correlated with respective NT titers (*p* < 0.0001, Spearman’s r = 0.92 and r = 0.80 for WT and Omicron, respectively). In addition, samples from individuals after primary Omicron infection were securely identified with the sVNT according to their distinctive nAb profiles (area under the curve = 0.99; sensitivity: 97.2%; specificity: 97.84%). Thus, when laborious live-virus NTs are not feasible, the novel sVNT we evaluated in this study may serve as an acceptable substitute for the serological identification of individuals with primary Omicron infection.

## 1. Introduction

With regard to neutralizing antibodies (nAbs), the Omicron variant of Severe Acute Respiratory Syndrome Corona Virus 2 (SARS-CoV-2) displays a significant antigenic distance to the ancestral wild-type (WT) virus [1,2,3,4,5]. This distance has severely reduced previously acquired and vaccine-derived humoral immunity against infection [1,2,3,6,7,8].

Thus, we and others applied variant-specific live-virus neutralization tests (NTs) and demonstrated that convalescents after primary Omicron infection display high titers of nAbs against the Omicron variant but only low titers against the WT and pre-Omicron variants [2,5,9,10]. Such Omicron-specific NT profiles contrast those after vaccinations and infections with the WT and previous variants [2,3,9,11].

While such distinct NT profiles may serologically identify convalescents after primary Omicron infection (e.g., in seroepidemiological surveys), commercial antibody assays that contain the Spike (S) protein of the ancestral WT as target antigen are reduced in their sensitivity to detect Omicron-S-specific antibodies [12,13].

For the characterization of neutralizing antibody profiles in large-scale seroepidemiological studies, live-virus NTs are not feasible, since they require biosafety level 3 laboratories and are time-consuming and laborious [14,15,16]. Therefore, even in the early phase of the pandemic, novel assays measuring the blocking effect of antibodies on the binding between RBD and ACE2 were developed, which proved to be a valuable surrogate for NTs [17,18].

This study evaluated a surrogate virus NT (sVNT) based on a commercial immunoassay that simultaneously measures the neutralizing activity against the WT (with the D614G mutation) and the Omicron variant. The first aim was to assess whether results from this sVNT correlated with the titers obtained from WT- and Omicron-specific live-virus NTs. The second aim was to evaluate whether the Omicron-adapted sVNT could detect antibodies specifically elicited upon primary Omicron infection in unvaccinated individuals, which are missed by antibody assays that contain the S protein (or the RBD) of the WT as the target antigen [12,13].

The sample cohort for this evaluation included 205 serum samples, which were quantified for the inhibition of the binding of angiotensin-converting enzyme 2 (ACE2) to the proteins of the variant-specific receptor-binding domains (RBDs). Corresponding live-virus NTs served as the reference. The samples were first selected for a wide range of neutralizing antibody titers after vaccinations or SARS-CoV-2 infection, and second, to compare samples after Omicron primary infection, and those after vaccination, Omicron vaccine breakthrough, and WT infection.

## 2. Materials and Methods

### 2.1. Samples

The study included an evaluation cohort of 205 samples (male: *n* = 68; female: *n* = 137; median age: 43.2 years; range: 4.6–96.8 years). For each person, only a single sample was included in the study. The samples were grouped according to the SARS-CoV-2 infection history and vaccination status of the donors.

Assignment as infection with the WT or the Omicron variant was based on the time point of infection. Thus, infections that occurred between March and December 2020 were considered WT infections (*n* = 43; during this period, only the WT variant circulated in Austria [19], and vaccinations were not yet available). Conversely, infections between January and April 2022 were considered Omicron infections (*n* = 71; Omicron variants BA.1 and BA.2 comprised over 99% of the cases during this period [19]). Out of those 71 samples from individuals with Omicron infections, 38 were obtained from individuals after probable primary infections with the Omicron variant who reported neither previous vaccination nor SARS-CoV-2 infection prior to the initial Omicron infection. The remaining 33 samples were considered probable Omicron breakthrough infections, as they occurred in subjects previously vaccinated two or three times (twice vaccinated: *n* = 5; three times vaccinated *n* = 28).

We also included samples from 91 individuals who were only mRNA-vaccinated without any report or evidence of a prior SARS-CoV-2 infection (i.e., nucleocapsid-specific antibodies were undetectable using Anti-SARS-CoV-2-NCP-ELISA; Euroimmun, Lübeck, Germany). Of those subjects, 55 samples were obtained after their second vaccination (7–9 months post-vaccination), and 36 were obtained 1 month after the third vaccination. A graphical overview of the inclusion criteria is provided as a flow chart in Appendix A, and detailed information on the cohort characteristics is provided in Appendix A.

### 2.2. Multivariant Surrogate Virus Neutralization Test

For this study, a multivariant sVNT was established by modifying the commercial SARS-CoV-2 VOC ViraChip^®^ IgG antibody assay (Viramed, Planegg, Germany). In brief, this microarray contains different SARS-CoV-2 proteins as target antigens (Spike 1, Spike 2, and Nucleocapsid protein) from the ancestral SARS-CoV-2 WT in addition to the RBD proteins from multiple SARS-CoV-2 variants (the ancestral WT strain, Delta, and Omicron variant BA.1) spatially separated into microspot triplets in each well of a 96-well plate.

This microarray was adapted to measure the antibody-mediated inhibition of ACE2 binding to the variant-specific RBD proteins. After incubation with the diluted serum samples, recombinant ACE2 bound to alkaline phosphatase (ACE2-AP; obtained from Viramed, Planegg, Germany) was added to the wells. Then, the reduction of the maximum color reaction catalyzed by the bound ACE2-AP in the presence of antibodies was measured and compared to a negative serum control containing no antibodies. Quantitative results were expressed as the percentage of ACE2-RBD binding inhibition calculated as 1-(Sample/Control) after subtracting the background (i.e., uninhibited color reaction = 0% binding inhibition).

### 2.3. Live-Virus Neutralization Test

The variant-specific live-virus NTs were conducted as previously described [2,20,21]. Briefly, the serum samples were incubated at 37 °C with 50–100 tissue culture infection dose 50 of either WT or BA.1 virus strains for one hour (WT with D614G mutation: GISAID accession number EPI_ISL_438123 [20,21]; BA.1: GISAID accession number EPI_ISL_9110894 [2]). Then, this mixture was applied to a monolayer of VeroE6 cells (ECACC 85020206). After 3–5 days, the NT titers were assessed using a microscope as the reciprocal dilution factor at which serum antibodies prevented a virus cytopathic effect. Serial dilutions ranged from 1:10 to 1:10240. NT titers ≥ 10 were considered positive.

### 2.4. Statistical Analyses

Data analyses were performed with GraphPad Prism 9.3.1. The correlation between WT- and Omicron-specific live-virus NT titers with respective levels of variant-specific RBD-ACE2 binding inhibition (including their ratios) was analyzed after log transformation using Spearman’s R and linear regression analyses. In addition, comparisons of WT vs. Omicron neutralizing antibodies were conducted for both assays using the Wilcoxon signed-rank test followed by Bonferroni correction. The alpha level was set to 0.05. Additionally, receiver operating characteristic (ROC) analyses were performed to analyze the correlation between quantitative levels of nAbs as assessed with the novel sVNT and variant-specific NTs and to estimate the respective sVNT level (in % ACE2-RBD inhibition) indicating WT- or Omicron-specific NT titers ≥10. Possible cutoff values were determined based on Youden’s J statistic. For assessing whether the sVNT could identify variant-specific neutralization patterns similarly to the live-virus NTs, ratios for the NTs were calculated by dividing the titer against Omicron by the respective titer against the WT (“NT-Ratio”). Analogously, “sVNT ratios” were calculated by dividing the values of % inhibition.

Thus, ratios greater than one represented a bias towards Omicron, whereas values below one represented a bias towards the WT. The sVNT ratios were only calculated for samples that were not oversaturated or not entirely negative (i.e., excluding those with both WT and Omicron inhibition of 100 or 0; *n* = 174). Finally, ROC analyses were performed to identify threshold values for sVNT ratios with the highest diagnostic accuracy to identify Omicron primary infections.

## 3. Results

### 3.1. Correlation of Variant-Specific nAbs as Assessed with sVNT and NTs

First, we analyzed the levels of nAbs against the WT and the Omicron variant in 205 samples from individuals after Omicron primary infection (*n* = 38), WT primary infection (*n* = 43), and vaccinated subjects with (*n* = 33) or without (*n* = 91) Omicron breakthrough infections. Levels of binding inhibition between ACE2 and the WT- and Omicron-specific RBD proteins, as assessed with the sVNT in %, were correlated with the nAb titers from the variant-specific live-virus NTs, which served as a reference.

As shown in Figure 1a,b, there was a highly significant correlation between results from the sVNT and the live-virus NTs for both variants (Spearman’s r = 0.92 and r = 0.80 for WT and Omicron, respectively, both *p* < 0.0001). Notably, high nAb titers, i.e., ≥80, against Omicron in the live-virus NT showed a better fit than lower titers, i.e., < 80 (Figure 1b).

Next, we performed ROC analyses to determine the optimal thresholds for the sVNT to detect any level of nAbs against the WT and the Omicron variant, using the limit of detection of the variant-specific live-virus NTs (any titer ≥ 10). The area under the curve (AUC) for detecting any level of nAbs using the sVNT was 0.98 for the WT and 0.88 for the Omicron variant (Appendix A). The optimal sVNT threshold for WT-specific ACE2-RBD binding inhibition was 7.5%, which provided a sensitivity rate of 95.4% and a specificity rate of 97.0% (Appendix A). For Omicron-specific ACE2-RBD binding inhibition, the sVNT displayed an overall sensitivity rate of 79.2% and a specificity rate of 92.9% using the optimal cutoff of 18.5% (Appendix A).

Then, we analyzed the effect that the additional use of the Omicron-specific RBD protein as a target antigen in the sVNT had on detecting nAbs in individuals with primary Omicron infection. Using the optimal cutoffs identified with ROC analysis, the sVNT displayed a sensitivity rate of 76.3% to detect nAbs in individuals after primary Omicron infection when Omicron-specific RBD-ACE2 binding inhibition was assessed. Notably, when binding inhibition between ACE2 and the WT-specific RBD protein was measured, the sensitivity of the sVNT to detect nAbs in individuals with primary Omicron infection was only 7.9% (Appendix A).

### 3.2. Profiles of Variant-Specific nAbs as Assessed with sVNT and Live-Virus NTs

Next, we investigated whether the cohorts differed regarding their neutralization profiles as assessed with the sVNT and the live-virus NTs. Indeed, subjects after primary Omicron infection showed significantly higher titers of nAbs against the Omicron variant than against the WT (Figure 1c). In contrast, all other cohorts had significantly stronger WT neutralizing activity (higher titers of nAbs against the WT) in the live-virus NTs than against the Omicron variant (all *p* < 0.0001; Figure 1c). Notably, the same pattern was found for the sVNT, with stronger WT reactivity than Omicron reactivity in samples from WT-infected individuals, whereas primary Omicron infection led to higher values in the blocking of the Omicron RBD-ACE2 binding (all *p* < 0.01; Figure 1d).

To compare the neutralization profiles obtained with the live-virus NTs and the variant-specific sVNTs in more detail, we calculated the ratios of the Omicron NT titer to the WT NT titer of each sample (“NT-Ratio”; ≥1: bias towards Omicron; <1: predominant for the WT), and similarly, the ratio of the Omicron sVNT to the respective WT sVNT result (“sVNT-Ratio”). Figure 2a displays the ratios for all subgroups.

Indeed, samples from individuals after primary Omicron infection displayed ratios with a bias towards the Omicron variant, and precisely, the same pattern was observed with the sVNT. As shown in Figure 2b, there was a highly significant correlation between the sVNT and NT ratios (r = 0.63, *p* < 0.0001). The ratios from both assays, live-virus NT and sVNT, thus concordantly identified nAb profiles weighted towards Omicron with a cluster in the upper-right quadrant of Figure 2b. Also, a ROC analysis confirmed that the sVNT ratio could serologically identify primary Omicron infections with high accuracy (area under the curve = 0.99, *p* < 0.005; optimal threshold: 1.25 sVNT ratio; sensitivity: 97.22%; specificity: 97.84%; Appendix A).

Finally, we compared the groups’ NT titers and sVNT levels (Appendix A). As shown in Appendix A, the statistically significant differences among the groups were similarly observed for NT titers and sVNT levels. As for WT neutralization, all pairwise comparisons differed significantly (*p* < 0.001 respectively) except for Omicron vaccine breakthroughs vs. three vaccinations, and two vaccinations vs. WT infections. Also, regarding Omicron neutralization, all cohorts differed significantly (*p* < 0.01 respectively) except for Omicron vaccine breakthroughs vs. three vaccinations, two vaccinations vs. WT infections, and Omicron primary infections vs. three vaccinations.

## 4. Discussion

In this study, we demonstrate that a surrogate virus neutralization test, adapted from a commercial immunoassay, can identify distinct profiles of neutralizing antibodies after primary Omicron infection with accuracy similar to that of variant-specific live-virus NTs.

With all its sub-variants, Omicron has become dominant worldwide, causing more and more breakthrough infections in vaccinated individuals and reinfections after previous infections with older variants [7]. Nevertheless, the serological identification of unvaccinated individuals after primary Omicron infection and the quantification of nAbs against multiple SARS-CoV-2 variants may still be relevant for seroepidemiological surveys focusing on population immunity, as has been demonstrated by Zaballa and colleagues [16].

However, due to the antigenic distance of the Omicron variant, commercial immunoassays that contain the Spike protein of the ancestral WT as the target antigen are significantly reduced in their sensitivity to detect antibodies in individuals after primary Omicron infection [12,13,22]. Indeed, this applies explicitly to commercial sVNTs, which still contain the RBD protein of the ancestral WT [12].

In our previous study, only 3–5% of the samples from convalescents after primary Omicron infection tested positive to commercial sVNTs, even though Omicron-specific nAbs were detectable with live-virus NTs [12]. Similarly, the sVNT we evaluated in this study would have only detected nAbs in 7.9% of individuals after primary Omicron infection if it only contained the WT-specific RBD protein as the target antigen.

Our recent data thus highlight that adding specific RBD proteins of Omicron variants as the target antigens in sVNTs may overcome their limitation. In this study, we adapted a commercial microarray to function as an Omicron-specific sVNT, restoring its sensitivity for the serodiagnosis of primary Omicron infections to 76.3%. Indeed, we additionally demonstrate that the adapted sVNT have specificity similar to that of live-virus NTs for differentiating antibodies formed against antigenically distinct SARS-CoV-2 variants, such as Omicron BA.1 and the WT.

Interestingly, the reactivity of the assay containing the Omicron-specific RBD was lower than the reactivity of the one with the WT-specific RBD, as indicated by a higher cutoff value for detecting the lowest level of nAbs that the respective live-virus NTs detected (cutoff values: 7.5% and 18.5% inhibition for WT and Omicron, respectively; limit of detection of the NTs: titers ≥10). Thus, the configuration or concentration of the Omicron-specific RBD protein used in the assay could be further improved and specifically adapted for the use as an sVNT.

Furthermore, the sVNT we evaluated in this study only contained the RBD protein of the BA.1 sub-variant, which certainly poses a limitation. Indeed, in the meantime, other sub-variants (e.g., BA.2, BA.5, XBB) with additional antigenic changes in the RBD have emerged [19]. Nonetheless, our primary finding that the diagnostic ability of the sVNT was restored after the RBD used as the target antigen was matched to the infecting variant might encourage the continuous adaption of these antibody assays to currently circulating variants.

As another limitation, we classified Omicron infections based on self-reporting by the infected individuals. However, for Omicron primary infections, specific profiles of NT titers have been reported (high against Omicron, low against the WT), which were present in all individuals from this group, which may again justify this classification [2,9,10,11,12].

Indeed, live-virus NTs are laborious and thus not feasible for large-scale seroprevalence studies. Therefore, sVNTs containing RBD proteins of multiple SARS-CoV-2 variants have been applied to gain insights into the levels of neutralizing antibodies against different SARS-CoV-2 variants, e.g., after natural infections and vaccinations or in seroepidemiological studies [14,16,23]. Data from our study now indicate that the application possibilities of sVNTs can be extended to the serological diagnosis of primary Omicron infections.

In this regard, our study highlights that sVNTs can identify patterns of nAbs against antigenically distinct variants of SARS-CoV-2, similarly to live-virus NTs. Although primary infections with SARS-CoV-2 might become increasingly rare in adults who have mostly been vaccinated, primary exposure to the Omicron variant will occur in most infants and children. As another possible application, sVNTs can quantify the increase in nAbs against distinct SARS-CoV-2 variants after bivalent vaccinations with the Spike proteins of the ancestral WT and different Omicron variants [21,22].

Similarly to previous studies using live-virus NTs alone, our comparative data obtained from live-virus NTs and the adapted sVNT concordantly indicate that Omicron breakthrough infections elicited higher NT titers against WT and Omicron as compared with vaccinated subjects without prior infections and unvaccinated individuals after WT or Omicron primary infections [2,9,10,11,24].

Furthermore, the neutralizing capabilities of the antibody response broadened after Omicron breakthrough infections (i.e., the observed WT/Omicron ratios were closer to 1) as compared with only vaccinated and WT-infected individuals, which can be attributed to the expansion of B memory cells that produce cross-reactive antibodies against shared epitopes [2,24,25,26,27].

Thus, not only after primary Omicron infections but also after vaccine breakthroughs, our sVNT detected similar patterns of nAbs as live-virus NTs, confirming its potential as a substitute assay. Expectedly, the sensitivity and specificity of the sVNT varied among the cohorts depending on their significantly different median nAb levels (Appendix A).

However, with the rapid emergence of new Omicron sub-variants, it will be essential to further adapt sVNTs, particularly when the antigenic distance among these variants increases. In the present study, the samples from individuals with primary Omicron infections mainly comprised BA.1-infected individuals (based on the infection dates), and this was concordant with the BA.1-specific RBD protein included in the assay. Thus, our study’s results prove that the diagnostic performance of sVNTs can be improved by matching the contained RBD proteins to the currently circulating variant, which can be extended to newer and upcoming SARS-CoV-2 variants.

## 5. Conclusions

In summary, this study demonstrates that an adapted sVNT that contains both RBD proteins, one of the WT and one of an Omicron variant, can specifically detect neutralizing antibody profiles that are distinctive for primary Omicron infections. This observation may thus be relevant for the further serodiagnosis of SARS-CoV-2 infections and for re-opening the potential of large-scale seroprevalence studies [16,28]. Indeed, the circulation of the Omicron variant is ongoing, and bivalent vaccines with the Omicron variant’s Spike protein are currently in use, calling for the further development and continuous adaptation of commercial immunoassays to the Omicron variants of SARS-CoV-2.

## Figures and Tables

**Figure 1 diagnostics-13-02278-f001:**
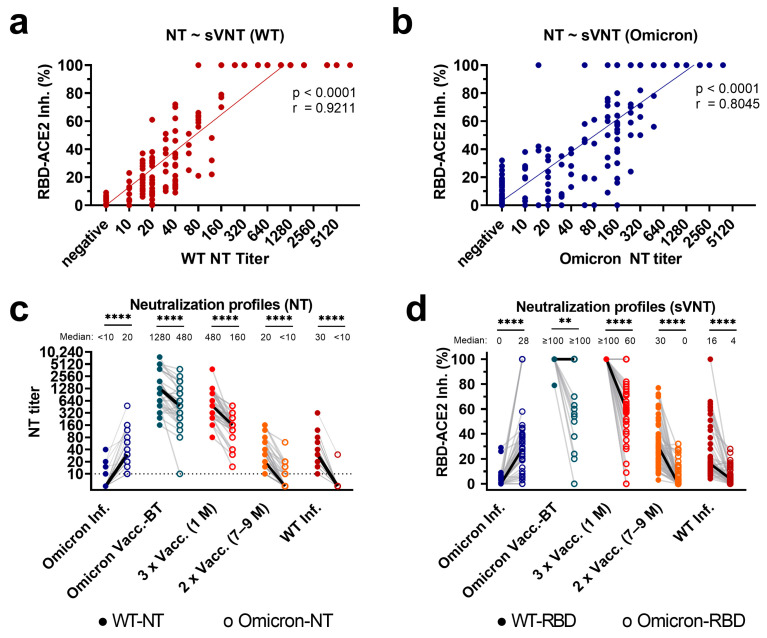
Correlation of sVNT results and live-virus NT titers, and comparison of neutralization profiles in the NT and sVNT. (**a**,**b**,**d**) The *y*-axis displays the binding inhibition between RBD and ACE2 in percentages (RBD-ACE2-Inh.%). (**a**,**b**) The *x*-axis and (**c**) the *y*-axis display neutralizing antibody titers as measured with (**a**) WT- and (**b**) Omicron BA.1-specific live-virus neutralization tests (NT titer). Each dot represents a single sample. (**c**,**d**) The *x*-axis displays the included cohorts: Omicron Inf.: Omicron primary infection in unvaccinated subjects (*n* = 38); Omicron Vacc.-BT: Omicron vaccine-breakthrough infection (*n* = 33); 3xVacc. (1 M): samples from individuals 1 month after the 3rd mRNA vaccination, no prior infection (*n* = 36); 2xVacc (7–9 M): samples from individuals 7–9 months after the 2nd mRNA vaccination, no prior infection (*n* = 55); WT Inf. Samples obtained from subjects after wild-type infection in the early phase of the pandemic (*n* = 43). (**c**,**d**) Full and empty circles display (**c**) neutralizing titers obtained with NTs and (**d**) RBD-ACE2 binding inhibition (in percent) obtained with the sVNT against the WT (full) and Omicron (empty), connected by a gray line for each sample. The solid black line indicates the median titers/percentages. (**a**,**b**) *p*-Values from linear regression on log-transformed NT data and Spearman’s r are indicated. (**c**,**d**) The difference in the respective WT and Omicron neutralizing activity was compared using Wilcoxon signed-rank tests followed by Bonferroni correction (**** *p* < 0.0001, ** *p* < 0.01).

**Figure 2 diagnostics-13-02278-f002:**
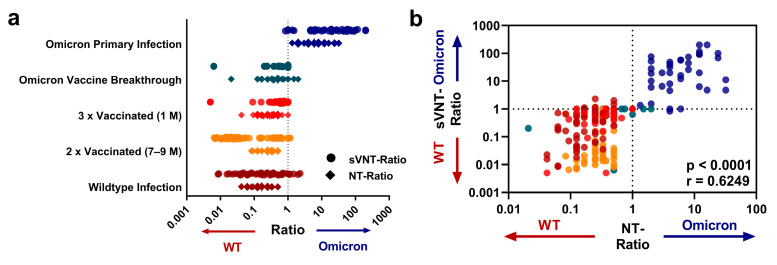
Ratio of Omicron to WT neutralizing activity of sVNT results and NT titers. (**a**) Log-transformed live-virus Omicron/WT NT titer ratios (diamonds) and Omicron/WT sVNT values (circles). A ratio greater than one (dashed line) indicates an Omicron-dominant neutralization profile, whereas ratios below one indicate a WT-dominant neutralization profile. (**b**) Correlation of sVNT value ratios and NT titer ratios. The color coding indicates the cohorts as presented in (**a**). The upper-right quadrant indicates samples in which both the NT and the sVNT ratios correspond to an Omicron-dominant profile, whereas samples in the lower left quadrant display a WT-dominant profile in both assays. The *p*-values displayed in the figure were obtained using linear regression, and Spearman’s r is indicated.

## Data Availability

Data from this study have not been publicly archived.

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
