# Peer review of "A Multivariant Surrogate Neutralization Assay Identifies Variant-Specific Neutralizing Antibody Profiles in Primary SARS-CoV-2 Omicron Infection"

_diagnostics, 2023, doi:10.3390/diagnostics13132278_

Round 1

Reviewer 1 Report

The authors write a brief report on how a new ACE2 inhibition assay correlates well with live virus neutralization results using a cohort of 205 individuals. The brief report is well written and, although previous studies have touched on the correlation between ACE2 inhibition assays and live virus neutralization, this study is more thorough than previous studies and includes both the WT and Omicron variants. As the authors state, these results are important as live virus assays are laborious and not feasible for very large cohorts. I have only minor comments. 

Specific comments

Line 98 - need to show how you calculate percent inhibition. This could also go in the statistics section. 

Line 32 - need to remove the MDPI instructions that have accidently been left in. 

line 169: need to explain a bit better what "RBD-ACE2 Inh.: Inhibition etc" and "Omicron Inf.: Omicron etc:" is. As in, need to link back to the graph and say "On the X axis..."

General comments

Please mention the current circulating variants in either the intro or the discussion and write a sentence on limitations of the paper and how you have not tested these currently circulating variants. 

Several studies have looked at live virus or pp assays and ACE2 competition assay correlations. Please ref these. (https://journals.asm.org/doi/full/10.1128/jcm.01533-20, https://journals.asm.org/doi/full/10.1128/msphere.00802-20)

Please add a flow chart of the cohort (see Walker et al, Viruses 2022 - supplementary).

Reviewer 2 Report

In this manuscript, Springer DN et al. describe the correlation of neutralizing antibodies determined by sVNT and variant-specific live virus neutralization tests (NTs) against WT and omicron BA.1. The author identified the cut-off of sVNT assay and determined the sVNT ratio and NT ratio to assess the pattern of sVNT, whether it is similar to the pattern of live-virus NTs. The findings presented in the manuscript are rather attractive since there is a problem with the accuracy of the current diagnostics and the new SARS-CoV-2 variants. However, there are some issues authors should take into account:

The statements in lines 132-134 should be removed.

1.) In lines 128-130, the authors mentioned, “ROC analyses were performed to identify threshold values for sVNT ratios with the highest diagnostic accuracy to identify Omicron primary infections.” It is not entirely clear why the authors specifically emphasize the results related to primary omicron infection and attempt to differentiate this cohort from the others. Could the author clarify this point? It would be acceptable if this study aims to adapt sVNT assay for detecting variant-specific neutralizing antibodies instead of using NTs for the entire population, which includes vaccinated individuals, infected individuals, and breakthrough infection.

2.) To help the reader, could the author identify the ROC analysis specifically for subgroups such as omicron breakthrough infection, vaccinated individuals without infection, and those with wild-type infection? Suppose the optimal cut-off values determined in this study were to be utilized in the future. What would be the sensitivity and specificity of this test to detect the neutralizing antibodies within different populations, including those with infection alone, vaccination alone, and hybrid immunity?

3.) The detail in lines 56-59 should be deleted. Due to this part being duplicated with the sample section in lines 67-77

This group should be explained that “How the author be so sure that there is no previous infection come before the Omicron period? If you didn’t follow them from the beginning, the author should add this as a limitation of the study. (Line 73: “38 was obtained from individuals after primary infections with the Omicron variant who reported neither previous vaccination nor SARS-CoV-2 infection before the initial Omicron infection.”)

Similarly, the group (n=91) without any report or evidence of a prior SARS-CoV-2 infection in line 78 should be mentioned how the author proved whether it is Anti- Nucleocapsid negative or not.

4.) The author should add the GMT for NT titer or the % inhibition from each group on top of Figures 1C and 1D to make it easier for the reader to compare NT levels.

5.) Line 171: Check the sample number of this group “samples one month after the 3rd mRNA vaccination, no 171 prior infections (n = 33)” It should be 36 or not

Line 219 typo “Omicron.”

-Line 226 typo “(12)”

6.) There are five various groups included in this study. The data comparing in FRNT or sVNT between groups such as omicron breakthrough and either no infected two or three doses do not be mentioned.

7.) It would be beneficial if the author provided a discussion on the topic of the Omicron vaccine breakthrough, specifically regarding the presence of neutralizing antibodies that target the wild-type (WT) strain at higher levels compared to the Omicron variant, as well as the similar response observed in individuals who received two to three vaccine doses without prior infection. Have you analyzed the sense & spec calculating from other groups (BT and vaccine alone).

8.) Fig 2a-b, the result of vaccine-BT and 2x & 3x vaccine alone does not be mentioned. It would be interesting to explore the reasons behind this association of nAb ratio with the WT strain in the omicron vaccine BT.

Reviewer 3 Report

Dear Author,

This is a well-planned study. The topic is highly relevant and may contribute to existing knowledge on surrogate Virus neutralization test (sVNT) which may specifically detect neutralizing antibody profiles that are distinctive for primary Omicron infections. However, the development and validation of such sVNT adapted from commercial immunoassays would require more extensive research and clinical trials to ensure its accuracy and reliability of measurements. 

Below, please find minor suggestions: 

Overall, this manuscript is well-written and the structure of the MS appears adequate and well divided in the different sections. 

MATERIALS and METHODS: 

While the sVNT may be effective in detecting neutralizing antibody profiles, the are different factors that may potentially affect accuracy and precision of the measurement results.  Some of these include: timing of sample collection, variability of immune response, cross-reactivity, and mutations in the virus. The dynamic of the immune response and neutralizing antibody production are very complex and may be influenced by comorbidities such as immunosuppression and autoimmune disorder, age, and disease severity. According to this please consider to completing the demographics characterization for all groups and add tables with “Clinical characteristics of study subjects” in different groups. I would recommend adding the timing of serum collection relative to the onset of COVID-19.

DISCUSSION and CONCLUSIONS

  I would recommend the Author to discuss the potential impact of the limitations when interpreting the study results.
